# Individual and Joint Associations Between Animal and Plant Protein Intakes with Impaired Fasting Glucose and Type 2 Diabetes in the Framingham Offspring Study

**DOI:** 10.3390/nu17010083

**Published:** 2024-12-28

**Authors:** R. Taylor Pickering, Ioanna Yiannakou, Laura Lara-Castor, M. Loring Bradlee, Martha R. Singer, Lynn L. Moore

**Affiliations:** 1Preventive Medicine and Epidemiology, Boston University School of Medicine, 72 East Concord Street, Boston, MA 02118, USA; ioannay@bu.edu (I.Y.);; 2Gerald J. and Dorothy R. Friedman School of Nutrition Science and Policy, Tufts University, Boston, MA 02111, USA

**Keywords:** diabetes, protein intake, animal protein, plant protein, diet, impaired fasting glucose, prospective cohort

## Abstract

Objectives: Given the considerable discrepancy in the literature regarding dietary protein and glucose homeostasis, we examined the prospective association between protein intake (total, animal, plant) and risk of type 2 diabetes mellitus or impaired fasting glucose (IFG). We also examined whether these associations differed by sex, body weight, or other risk factors. Methods: We included 1423 subjects, aged ≥ 30 years, in the Framingham Offspring Study cohort. Three-day dietary records at exams 3 and 5 were used to average protein intake and then adjusted for body weight residuals. Cox proportional hazard models were used to estimate hazard ratios (HR), adjusting for anthropometric, demographic, and lifestyle factors over ~16 years of follow-up. Results: Subjects with the highest total protein intakes (≥100 g men; ≥85 g women) had a 31% lower risk of type 2 diabetes/IFG (95% CI: 0.54, 0.87). The highest (vs. lowest) category of intake of animal protein was associated with a 32% lower risk of diabetes/IFG (95% CI: 0.55, 0.83), whereas plant protein was not. Beneficial trends of total protein, especially animal, were stronger for women (HR: 0.61; 95% CI: 0.42, 0.87) than for men (HR: 0.82; 95% CI 0.58, 1.15). Subjects with lower BMI who consumed more protein had the lowest risks of diabetes/IFG. Conclusions: Overall, in this prospective study, higher intake of total dietary protein, including the consumption of animal protein, particularly among individuals with lower BMI and higher physical activity levels, was inversely associated with risk of incident type 2 diabetes and IFG.

## 1. Introduction

Current guidelines for Americans to prevent type 2 diabetes mellitus recommend weight management therapies through lifestyle approaches such as diet and physical activity, especially for those who are obese or overweight [1]. A meta-analysis, including 21 controlled trials, showed that participants consuming a high-protein and low-fat diet had 0.79 kg greater weight loss than those consuming an isocaloric energy-restricted low-protein and low-fat diet. Larger weight reductions were observed in trials of longer duration [2]. The connection of a higher protein diet to weight loss could be due by a number of mechanisms including reduction in appetite, diet-induced thermogenesis, or preservation of lean mass which may lead to higher basal metabolic rate, lowering weight and fat mass [2,3]. Since the trials in the above meta-analysis compared calorie-matched dietary groups, the difference in weight reduction was small but still clinically important given that prior evidence showed that weight reductions as small as 1.0 kg in individuals with obesity could lead to a 16% lower risk of type 2 diabetes [4]. Taking these data together, a diet high in protein content could potentially serve as a lifestyle approach to reducing obesity and thereby preventing impaired fasting glucose and, ultimately, type 2 diabetes.

To date, several epidemiological studies have examined the association between dietary protein intake and the risk of type 2 diabetes, but the results are largely inconclusive, perhaps due to a number of factors including differences in study design and difficult to control participant characteristics (e.g., socioeconomic status and adiposity). While some prospective cohort studies have reported an increased diabetes risk associated with higher total protein intakes [5,6,7,8,9], several short-term, randomized clinical trials have found either no effect [3,10] or a protective effect [11,12,13]. Interestingly, Sluik and colleagues harmonized data from four cohorts (three European, one Canadian), with a total population of 78,851 participants who were free of prevalent type 2 diabetes, and found that a higher total protein intake was associated with lower incidences of pre-diabetes and diabetes [14]. In this and other prior studies [5,7,8], associations were attenuated after adjusting for body mass index (BMI) and/or adiposity measures, as well as attempting to control for other factors associated with protein intake such as income and physical activity. While BMI or other measures of body fat may be confounders or causal intermediates in the pathway to diabetes, they may also be modifiers of the effects of protein on diabetes risk. However, few studies have stratified the associations between protein intake and type 2 diabetes risk by BMI [5,7,8]. Additional studies are needed to elucidate the role of BMI in the association between protein intake and type 2 diabetes.

The source of dietary protein (animal vs. plant) has been shown to have variable effects on the risk of type 2 diabetes. While some studies have found no association between animal or plant protein intake and risk of type 2 diabetes [7,15], others have found higher risks associated with animal protein intake and lower risks associated with higher plant protein intake (or by modeling replacement of animal protein with plant proteins [6,9]). Further, some studies have found the highest risks to be associated with the consumption of red or processed meats [15,16] rather than other sources of animal protein such as dairy [17]. Therefore, it is still unclear whether the effects of protein on type 2 diabetes are due to protein per se or due to other nutrients found in the protein-rich foods or dietary patterns that may accompany protein-rich foods.

Given the importance of lifestyle guidance in managing the risk of type 2 diabetes and the current uncertainty in the literature regarding dietary protein and glucose homeostasis, it is evident that additional data on the effects of protein from various sources are needed. Thus, our objective was to examine the associations of total, animal, and plant protein intakes on the risk of type 2 diabetes or impaired fasting glucose (IFG) in the Framingham Offspring Study. We also evaluated whether these relations were modified by BMI or other dietary or lifestyle factors.

## 2. Materials and Methods

### 2.1. Study Design and Participants

The Framingham Offspring Study began in 1972 and enrolled a total of 5124 offspring (and their spouses) of participants in the original Framingham Heart Study [18]. Data related to cardiometabolic risk factors, psychosocial factors, lifestyle habits, and medical history were collected at examination visits at approximately four-year intervals. Dietary information was derived from dietary records collected at exams 3 (1983–1988) and 5 (1991–1995); thus, the baseline for the current analyses is considered at the end of the dietary exposure period (exam 5). Figure 1 shows the timing of the relevant variables used in the current study. Follow-up was through exam 8 (2005–2008).

For these analyses, we included those who were 30–54 years of age and had complete and valid dietary information at baseline including plausible energy intakes (men reporting between 1200 and 4000 kilocalories/day (kcals) and women reporting energy intakes 1000 ≤ kcals/d ≤ 3500) (n = 1744). We excluded those subjects with prevalent impaired fasting glucose or type 1 or type 2 diabetes (n = 296) at baseline (i.e., exam 5 when follow-up started) and those missing covariates included in the final model (n = 25), resulting in a final sample size of 1423.

### 2.2. Dietary Assessment

Two sets of 3-day diet records (6 days in total) were collected during exam cycles 3 and 5 using standardized protocols. Two-dimensional food models were used to aid in the estimation of portion sizes. Data from the dietary records were analyzed for nutrient content using the Nutrition Data System of the University of Minnesota [19]. The Nutrition Data System food codes were linked with United States Department of Agriculture food codes [20] to derive standardized servings [21]. Usual protein intake from all sources as well as separately from animal (e.g., milk, cheese, eggs, beef, pork, chicken, fish) and plant (e.g., legumes, nuts, seeds, whole grains) sources was estimated as the mean from all six days of dietary records.

### 2.3. Outcome Measurement

Mean fasting glucose was assessed at the baseline for these analyses (exam 5) and the following subsequent exams (exams 6 (1995–1998), 7 (1998–2001), and 8 (2005–2008)). Blood specimens were collected in a standardized manner following a 12 h fast and then stored at −80 degrees centigrade as previously described [22]. IFG was defined as having a fasting glucose between 100 and 125 mg/dL. Type 2 diabetes was defined as having a fasting glucose ≥ 126 mg/dL, non-fasting glucose ≥200 mg/dL, or current treatment with oral antidiabetic medication or insulin [23].

### 2.4. Potential Confounders

A wide range of potential confounders was regularly measured in the Framingham Offspring Study. Factors explored as potential confounders at baseline in this analysis included the following: age, sex (except in sex-stratified analyses), education, physical activity, height, BMI, systolic and diastolic blood pressure, prevalent high blood pressure, total energy intake, alcohol intake, and weight-adjusted intakes of carbohydrates, total fat, and saturated fat. Self-reported education was categorized as high school or less vs. more than high school. Weight was measured with a standard balance beam scale and height with a standard stadiometer. BMI at the baseline exam was calculated at the exam-specific weight (kg) divided by average (up to age 60) adult height (m) squared. Information on self-reported smoking was updated at each exam and expressed as cigarettes smoked per day. Physical activity was assessed between 1987 and 1991 by asking each subject to report the number of hours/day spent in sleep, sedentary, light, moderate, and vigorous activities in 24 h. A physical activity index was created by summing the number of hours spent in moderate and vigorous activities and multiplying it by a weighted estimate of energy expenditure (oxygen consumption) required for activities of that level of intensity as has been previously described [24].

### 2.5. Statistical Analyses

Protein and other macronutrient intakes were estimated using residuals from a linear regression model adjusting for body weight. Each participant’s macronutrient residual was added to the median intake level for that nutrient for the overall group. The following sex-specific categories of protein intake were chosen based on sensitivity analyses combined with power considerations: low (<75 g/day for men and <65 g/day for women), moderate (75 g/day to <100 g/day for men, and 65 g/day to <85 g/day for women) and high (≥100 g/day for men and ≥85 g/day for women). Animal protein intake was classified as follows: low (<55 g/day for men and <40 g/day for women), moderate (55 g/day to <65 g/day for men and 40 g/day to <55 g/day for women), and high (≥65 g for men and ≥55 g/day for women). Plant protein intake was similar for men and women and was classified as follows for both sexes: low (<20 g/day), moderate (20 g/day to <25 g/day), and high (≥25 g/day).

Cox proportional hazard models (total and sex-specific) were conducted to assess the hazard ratio (HR) for incident type 2 diabetes or IFG. The proportional hazards assumptions were tested in all models, and no violations of the assumptions were found. Person-years of follow-up for each participant were calculated as the time from baseline dietary assessment to the first of the following events: incident type 2 diabetes or IFG, lost to follow-up, death, or end of exam 8 (2005–2008).

Effect modification analyses were used to examine whether the association between protein intakes and diabetes risk might depend on the level of other risk factors, including physical activity, BMI, or scores on the 2015 Healthy Eating Index (HEI-2015) [25] as an indicator of a generally healthy diet. The HEI used in these analyses was modified to exclude the total protein foods component of the index. Effect modification was assessed on an additive risk ratio scale [26]. Sensitivity analyses were used to dichotomize each protein intake and possible effect modifier to optimize statistical power. For example, total protein intake was dichotomized to enhance statistical power as follows: lower (<75 g/day for men and <70 g/day for women) vs. higher (≥75 g/day for men and ≥70 g/day for women). Animal protein intakes were categorized as lower (<55 g/day for men and <45 g/day for women) vs. higher (≥55 g for men and ≥45 g for women). Finally, plant protein intakes for both men and women were categorized as lower (<20 g/day) vs. higher (≥20 g/day). Lower vs. higher BMI was classified as <28 kg/m^2^ for men and <25 kg/m^2^ for women vs. ≥28 kg/m^2^ for men and ≥25 kg/m^2^ for women). Low vs. higher physical activity levels were classified as <9.6 vs. ≥9.6 METs/h. Finally lower vs. higher HEI scores were classified as <50 vs. ≥50 scores. Next, we cross-classified the categories of the protein intakes with the categories of the effect modifier yielding four possible combinations. For example, for HEI we classified subjects into one of the following four categories: (1) lower protein intake score and lower HEI scores (<50 scores) (referent group = hypothesized to have the highest risk), (2) higher protein intake and lower HEI scores, (3) lower protein intake and higher HEI scores (≥50 scores), and (4) higher protein intakes and higher HEI scores. Cox proportional regression models were used to compute the HRs in each category. The difference in HR estimates between categories 2 vs. 1 indicates whether there is a protective association of a higher protein intake alone while the comparison of category 3 vs. 1 estimates the effect of higher HEI scores alone. Finally, the HR for category 4 vs. 1 shows the effect of having both preventive factors together (vs. neither).

Only factors that were found to be confounders (as defined by a 5% change or more in the overall effect estimates) of the relation between total, animal, or plant protein intake and type 2 diabetes/IFG were included in the final models. Final multivariable models were adjusted for sex, age (years), education (high school or less vs. some college or more), physical activity, cigarettes (number per day), height, energy intake, and weight-adjusted carbohydrate intake.

## 3. Results

Baseline characteristics of participants according to total protein intake categories (low, moderate, and high) are presented in Table 1. Participants in the highest total protein intake group consumed approximately 106 weight-adjusted grams (g) of protein per day (1.48 g/kg of body weight) compared with 58 g of protein in the lowest total protein group (0.78 g/kg per day). Higher protein intake was associated with lower intakes of carbohydrates and slightly higher intakes of fat. Further, those in the highest total protein intake group were slightly younger, had a lower BMI, and were more active than those consuming less protein.

Table 2 shows the relationship between total, animal, and plant protein intakes and the risk of developing type 2 diabetes or IFG. Among the overall population, those with moderate total protein intakes (vs. lower) had a 21% lower risk (HR: 0.79; 95% CI: 0.66, 0.94) of type 2 diabetes/IFG while those with the highest (vs. lowest) intakes had 31% lower risks (HR: 0.69; 95% Cl: 0.54, 0.87). Moderate and higher (vs. lower) animal protein intakes were associated with statistically significant 29% and 32% lower risks, respectively, of type 2 diabetes/IFG. There were no associations between plant protein intake and type 2 diabetes/IFG. Sex-stratified analyses showed that the associations between protein intakes and type 2 diabetes/IFG were stronger in women than in men. Women with moderate and higher intakes of both total protein and animal protein had lower risks of these outcomes. For example, the highest consumption of total protein was linked with a 39% lower risk (HR: 0.61; 95% Cl: 0.42, 0.87) of type 2 diabetes/IFG among women but only an 18% (non-statistically significant) lower risk (HR: 0.82; 95% Cl: 0.58, 1.15) in men. Similarly, women with the highest animal protein intakes had a 41% lower risk of type 2 diabetes/IFG, while men had a 24% lower risk. Again, there was no association between plant protein consumption and risk of type 2 diabetes/IFG in either men or women.

Figure 2 demonstrates associations of total, animal, and plant protein on type 2 diabetes/IFG modified by level of physical activity, BMI, or adherence to scores on the HEI-2015. Higher total protein intake combined with higher physical activity levels, or a lower BMI was associated with lower risks of type 2 diabetes/IFG, but these combined impacts were not additive. For example, higher total protein intake among participants with a lower BMI was associated with a 46% lower risk of type 2 diabetes/IFG, while higher protein intake alone was associated with an 18% lower risk and BMI alone with a 39% lower risk. Similarly, those with higher animal protein intakes who also had a lower BMI had a 50% lower risk (HR: 0.50; 95% CI: 0.40, 0.62) of type 2 diabetes/IFG compared with 22% and 33% lower risks for higher animal protein and lower BMI levels alone, respectively.

The associations of plant protein on diabetes risk in these stratified analyses (Figure 2) were weaker than those for animal protein or total protein. However, we did observe that high plant protein intake combined with higher physical activity levels was associated with a statistically significant 25% reduction in type 2 diabetes/IFG risk (95% Cl: 0.57, 0.99), while neither higher plant protein nor higher activity alone was associated with lower risk. Finally, we found that higher HEI scores alone were not associated with the risk of type 2 diabetes/IFG and did not modify the associations of protein intake in these analyses.

In Figure 3, we examined possible sex-specific differences in the independent and combined effects of dietary protein intakes and other risk factors. Both higher physical activity and a lower BMI, especially in combination with higher intakes of total protein or animal protein, were associated with lower risks of type 2 diabetes/IFG. In general, the HEI-2015 scores were not independently associated with the risk of type 2 diabetes/IFG in either men or women. For example, among men, higher intakes of animal protein were linked with statistically significantly lower risks of diabetes/IFG regardless of their HEI-2015 scores. Among women, those with higher intakes of animal protein had approximately 36–37% lower risks of diabetes/IFG regardless of scores on the HEI. However, the HEI score itself was not associated with diabetes/IFG risk.

In these analyses (Figure 3, Appendix A), the associations of animal protein intake were generally stronger and more precise than those for plant protein. BMI itself, especially in women, was a determinant of the risk of type 2 diabetes and IFG. Among men with lower animal protein intakes, a BMI < 28 kg/m^2^ (vs. a higher BMI) was associated with a 27% lower risk (95% CI: 0.52, 1.03) of type 2 diabetes/IFG. Those men with both a BMI < 28 kg/m^2^ and higher protein intakes (vs. those with a higher BMI and lower protein intakes) had a 38% lower risk (95%CI: 0.45, 0.84) of type 2 diabetes/IFG. Thus, the combined impacts of a lower BMI and higher intake of animal protein were stronger than those associated with either factor alone. In stratified analyses of plant protein intake among men, we found that a BMI < 28 kg/m^2^ (vs. a higher BMI) was associated with a 39% lower risk of type 2 diabetes/IFG. Adding higher plant protein intakes did not enhance the risk reduction associated with a lower BMI. Among women, the combined effect of a BMI < 25 kg/m^2^ and higher animal protein intake (vs. BMI ≥ 25 kg/m^2^ and lower animal protein intake) was associated with a 59% lower risk of diabetes/IFG, showing that these combined effects were stronger than either the associations of higher animal protein intake alone (a non-statistically significant 23% risk reduction) or a lower BMI alone (a statistically significant 39% risk reduction). For plant protein, women with a BMI < 25 kg/m^2^ who consumed more plant protein (vs. women with a higher BMI who consumed less) had a 49% lower risk (95% CI: 0.37, 0.71) of diabetes/IFG. However, most of this lower risk was likely attributable to the lower BMI because those women with a lower BMI alone (but lower plant protein intake) still had a 42% lower risk of diabetes/IFG.

## 4. Discussion

In this prospective study of adults with up to 16 years of follow-up, we found a lower risk of type 2 diabetes and IFG associated with total protein consumption, especially among women. In both women and men, there was an inverse association between animal protein in the diet and risk of type 2 diabetes/IFG. However, there was no statistically significant overall association between plant protein consumption and type 2 diabetes/IFG risk in either men or women. We further examined the independent and combined impacts of dietary protein from different sources and other risk factors, including physical activity, BMI, and diet quality as measured by HEI-2015 scores. These analyses showed that the lowest risk of type 2 diabetes/IFG risk was seen in those with low BMI and those who consumed more protein, especially animal protein, in women. Lastly, participants who were more physically active and who consumed more total protein had lower risks of diabetes and IFG.

Our study contrasts with those of some other studies that found positive associations between total protein intake and risk of type 2 diabetes. Data from the Nurses Health Study (NHS) and the Health Professionals Follow-up Study (HPFS) with more than 15,000 cases of type 2 diabetes found that participants in the highest (vs. lowest) quintile of energy-adjusted total protein intakes had a 7% higher risk of type 2 diabetes [5]. The highest quintile of animal protein consumption was associated with a 13% higher diabetes risk while plant protein intake was associated with a 9% lower risk. A recent meta-analysis of seven prospective cohorts estimated that higher intakes of animal protein (as defined variably across studies) were associated with a 13% higher risk of type 2 diabetes [6]. However, these results ranged from a relative risk of 0.64 to 1.44 across studies and were most heavily weighted by the above results in the NHS and HPFS cohorts. Further stratification of the pooled results by country of origin showed that higher intakes of animal protein were protective against type 2 diabetes in Asian populations but not in Australian, U.S., or European populations. In contrast, an analysis of data from the PREVIEW Project, which included three European and one Canadian prospective cohorts, found 16% and 51% lower risks of pre-diabetes and diabetes, respectively, associated with higher total protein intake [14]. Higher intakes of both animal and plant protein were associated with lower risks of incident diabetes.

There are several possible explanations for the variability in results across studies and between the findings in the current study and some prior studies. First, the dietary patterns of participants in these different study populations are highly variable. There are also differences in the methods of dietary assessment and methods used to adjust dietary protein for overall body size or caloric intake. There is no universal consensus on how protein intake should account for these differences in energy intake or body size. Many studies express protein intake as a percent of calories [5,9], while others adjust protein intake for energy intake using the residual method [7,15,27]. Other investigators express protein intake as the number of grams of protein consumed per kg of body weight [14]. Since protein requirements may be more closely linked with lean body mass than fat mass, some have proposed expressing intake as grams of protein per kg of ideal body weight [28]. In the current study, we adjusted protein intake for actual body weight using the residual method and further adjusted for height in the multivariable models. These different approaches to determining protein intake may yield different study results. Further, the estimation of total energy intake and other nutrients may differ across studies according to participant characteristics, the dietary assessment method used, and the types of foods eaten. For example, energy intake might be underreported due to the omission of calorie-dense, protein-poor snacks consumed between meals [29]. This differential reporting impacts estimated energy intake and could bias results when protein intake is expressed as a function of total energy intake. Further, the majority of studies have used food frequency questionnaires to assess dietary intakes, and since energy intake is not measured well with a food frequency questionnaire, the energy-adjusted intakes of dietary protein will also be subject to considerable error (which may bias the results). In our analysis, we assessed dietary intake using food diaries and then adjusted the grams of protein intake for body size in an attempt to minimize errors in the estimation of protein intake.

We found some evidence suggesting that BMI modifies the associations of animal protein intake on type 2 diabetes/IFG. Very few studies have examined the modifying role of BMI on protein intake with respect to diabetes-related outcomes. In the PREVIEW Project, the lower risks of type 2 diabetes associated with protein intake did not differ across categories of BMI [14], while in the Dutch European Prospective Investigation into Cancer and Nutrition, investigators found no association between protein and diabetes risk among obese individuals (BMI ≥ 30 kg/m^2^) but an increased risk among the non-obese [8]. In the current Framingham cohort, we found that participants with higher intakes of animal protein tended to have lower risks of type 2 diabetes/IFG regardless of body weight, although the beneficial effects of higher animal protein intakes were stronger and only reached statistical significance among individuals with lower BMI. It is possible that some of the beneficial effects of dietary protein on diabetes risk may be explained by a reduction in the acquisition of age-related fat mass. Previous data from the Women’s Health Study, for example, showed that women with lower adiposity in the highest quintile of dairy intake (rich in animal protein) gained less weight over 11 years of follow-up than those in the lowest quintile of intake [30]. These data may help to explain some of more beneficial effects in women in this study, as the source of animal protein (e.g., dairy vs. meat) may exhibit sex-specific differences.

It is also possible that the source of protein has variable effects on weight, weight loss, BMI, and glucose homeostasis. In one 12-week parallel design controlled clinical trial, 25 overweight and obese individuals who were randomized to a group supplemented with whey protein had no statistically significant reductions in body weight but showed statistically significant decreases in insulin levels and insulin resistance from baseline to the end of follow-up [13]. However, those in the glucose control group (n = 25) or a casein-supplemented group (n = 20) showed modest levels of weight loss but increases in insulin levels and insulin resistance from baseline to the end of follow-up. These whey-protein-related improvements in insulin sensitivity were not explained by a loss of fat mass or decreased waist circumference. It is, therefore, possible that high protein diets may be beneficial through weight-independent mechanisms, such as lowering of glycemic load or insulinotropic effects. Although evidence is very limited among healthy individuals, milk proteins, especially whey [31] and branched-chain amino acids [32], have been shown to promote postprandial insulin secretion through both direct effects on pancreatic beta cells and increased incretin production that enhances insulin secretion [33]. This higher insulin response to high protein intake could promote greater glucose control in healthy individuals [32] as well as those with type 2 diabetes [34]. However, a recent meta-analysis of 15 randomized controlled trials of more than 12 months duration studied the long-term association of diets high in total protein (>25% of energy) and found neither a positive nor a negative impact on glycemic control, and no weight loss compared with diets low in protein content (10–15% of energy from protein) in both healthy and insulin-resistant subjects [2].

Physical activity is well-recognized as an important factor in diabetes prevention. This study shows beneficial effects of both dietary protein and physical activity in reducing type 2 diabetes/IFG risk, with the greatest effect being the combination of the two. In line with our results, a randomized controlled trial showed that adequate or moderately high amounts of dietary protein along with resistance training led to improvements in oral glucose tolerance and insulin signaling in skeletal muscle without changes in body weight [35].

There are several important strengths of the current study. Its prospective design and repeated measures data allow for more robust tracing of disease progression. In addition, two sets of 3-day dietary records, a gold-standard approach for estimating dietary intake [36], were averaged to estimate usual protein consumption prior to the start of follow-up, thus providing a more accurate and precise measure of the exposure. The dietary records were collected following carefully standardized protocols with extensive debriefing, enabling us to determine the separate intakes of animal and plant proteins. Nonetheless, dietary intakes were self-reported, and thus are subject to some degree of error in reporting some foods. Finally, the 16 years of follow-up for the incidence of diabetes or the presence of IFG is a strength of this study that minimizes the possibility of reverse causality as an explanation for these findings. In addition, the diabetes outcomes were collected and adjudicated as were the potential confounding and effect-modifying variables. The careful prospective collection of these many factors increases the likelihood of deriving unbiased estimates of associations in this study.

One key limitation of the study is the relatively low plant protein intake (mean ± s.d.: 22 g/d ± 8) in this population. Thus, we were not able to compare equivalent amounts of dietary protein from animal and plant sources. Nonetheless, the fact that animal protein was the predominant protein source does indicate that in this study, at least, higher intakes of animal protein had no adverse effects on the long-term risk of type 2 diabetes or IFG. Further, dietary records collected two times over 12 years likely do not fully represent dietary variety or consistency within our population and often result in under-reporting of foods deemed “less healthy” in a body-size-dependent manner [29,37]. In this study, there are relatively few cases of incident type 2 diabetes (183), thus we could not examine separate effects of dietary protein on diabetes and IFG. Fasting glucose is variable over time, as such some individuals did revert to normal fasting glucose levels after being classified as having IFG. While present, this occurred only in 1.9–5.2% of cases, depending on which visit IFG was observed. As with all observational studies, residual confounding may be present. Further, participants in this study were exclusively Caucasian and highly educated (66.7% of the study population had a high school degree or beyond), so these results may not be representative of risks in a multiethnic diverse population with different levels of education and socioeconomic status. Lastly, we acknowledge that the dietary patterns at the time when these data were collected may not reflect current dietary habits and intakes.

## 5. Conclusions

Our study provides important evidence that total protein consumption, including the consumption of animal protein, was associated with a lower long-term risk of diabetes among middle-aged adults, and these associations were stronger in women. These observations indicated higher protein intake may help prevent the development of dysregulated glucose homeostasis; however, well-controlled clinical trials assessing diet patterns with more balanced intakes of animal and plant protein are needed to determine whether comparable amounts of animal and plant proteins yield similar changes in these important metabolic outcomes.

## Figures and Tables

**Figure 1 nutrients-17-00083-f001:**
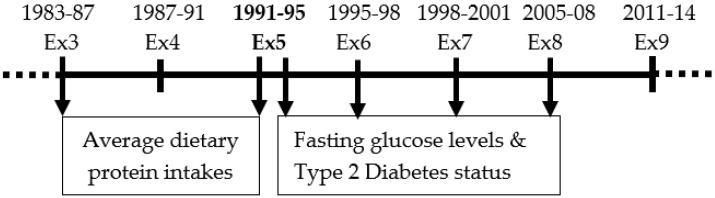
Timeframe and data collection for variables utilized in the current analyses from the Framingham Offspring Study. Arrows indicate when measures were taken. Ex, Exam.

**Figure 2 nutrients-17-00083-f002:**
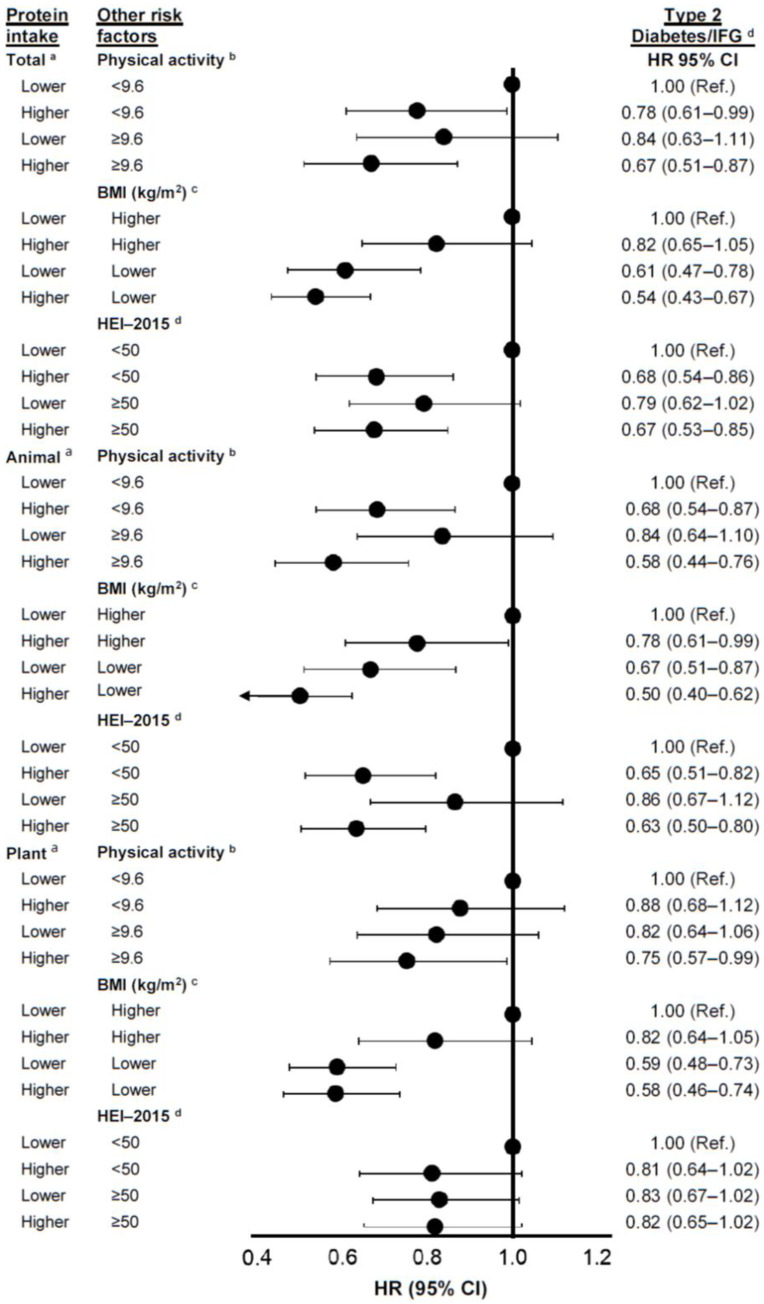
Modification of the associations of dietary protein (total, animal, and plant) on type 2 diabetes or impaired fasting glucose risk by physical activity, body mass index, and the Healthy Eating Index-2015 (excluding all the protein components) in the Framingham Offspring Study. Models were adjusted for age, sex, education level, physical activity, smoking cigarettes per day, height, energy intake, and weight-adjusted carbohydrate intake. Models assessing effect modification by physical activity did not include activity in the model. Models for animal protein were also adjusted for plant protein intakes and vice versa. ^a^ Lower vs. higher total protein intakes: <75 g/d for men and <65 g/d for women vs. ≥75 g/d for men and ≥65 g/d for women. Lower vs. higher animal protein intakes: <55 g/d for men and <45 g/d for women vs. ≥55 g/d for men and ≥45 g/d for women. Lower vs. higher plant protein intakes: <20 g/d vs. ≥20 g/d for both men and women. ^b^ Physical activity units were METs/day. ^c^ Lower vs. higher BMI: <28 kg/m^2^ for men and <25 kg/m^2^ for women vs. ≥28 kg/m^2^ for men and ≥25 kg/m^2^ for women. ^d^ HEI-2015 score excluding all protein components. BMI, body mass index; g/d, grams per day; HEI, Healthy Eating Index; kg/m^2^, kilograms per meter squared; METs/d, metabolic equivalents per day; Type 2 diabetes/IFG, type 2 diabetes mellitus or impaired fasting glucose.

**Figure 3 nutrients-17-00083-f003:**
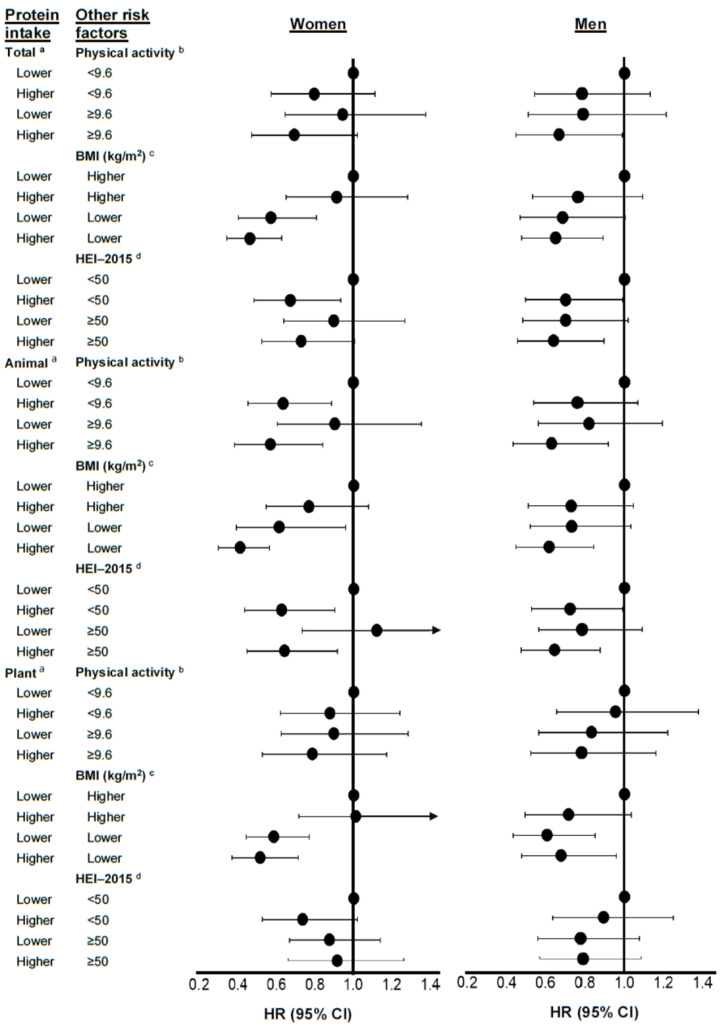
Modification of the associations of dietary protein (total, animal, and plant) on type 2 diabetes or impaired fasting glucose risk by physical activity, body mass index, and the Healthy Eating Index-2015 (excluding all the protein components) in women and men of the Framingham Offspring Study. Models were adjusted for age, sex, education level, physical activity, smoking cigarettes per day, height, energy intake, and weight-adjusted carbohydrate intake. Models assessing effect modification by physical activity did not include activity in the model. Models for animal protein were also adjusted for plant protein intakes and vice versa. Hazard ratios and confidence limits can be found in Appendix A. ^a^ Lower vs. higher total protein intakes: <75 g/d for men and <65 g/d for women vs. ≥75 g/d for men and ≥65 g/d for women. Lower vs. higher animal protein intakes: <55 g/d for men and <45 g/d for women vs. ≥55 g/d for men and ≥45 g/d for women. Lower vs. higher plant protein intakes: <20 g/d vs. ≥20 g/d for both men and women. ^b^ Physical activity units were METs/day. ^c^ Lower vs. higher BMI: <28 kg/m^2^ for men and <25 kg/m^2^ for women vs. ≥28 kg/m^2^ for men and ≥25 kg/m^2^ for women. ^d^ HEI-2015 score excluding all protein components. BMI, body mass index; g/d, grams per day; HEI, Healthy Eating Index; kg/m^2^, kilograms per meter squared; METs/d, metabolic equivalents per day.

**Table 1 nutrients-17-00083-t001:** Baseline characteristics of participants according to total protein intake in the Framingham Offspring Study.

	Total Protein Intake ^1^		
Baseline Characteristics	Low ^2^ (n = 399)	Moderate ^2^ (n = 646)	High ^2^ (n = 378)
Age (years)	44.5 ± 6.2	43.6 ± 6.5	43.2 ± 6.4
BMI (kg/m^2^)			
All participants	26.8 ± 4.9	25.2 ± 4.0	24.7 ± 3.5
Women	25.8 ± 5.1	24.2 ± 4.1	23.4 ± 3.0
Men	28.3 ± 4.1	26.6 ± 3.4	26.2 ± 3.5
Physical activity index (METs/d)	12.0 ± 7.8	12.1 ± 7.6	12.6 ± 8.1
Number of cigarettes per day	7.1 ± 12.8	5.0 ± 10.9	5.2 ± 11.7
Current smoker (n, %)	121 (30.3)	149 (23.0)	85 (22.4)
Energy intake (kcals)	1591 ± 405	1900 ± 451	2383 ± 575
Protein intake			
Weight-adjusted protein (g)	57.9 ± 9.2	79.7 ± 8.6	105.6 ± 15.1
Animal protein (g)	38.9 ± 8.7	56.8 ± 9.1	77.0 ± 14.3
Plant protein (g)	18.0 ± 5.4	21.7 ± 6.6	27.3 ± 9.6
Protein (% energy)	15.1 ± 2.9	16.9 ± 3.2	17.9 ± 3.1
Protein (g/kg/day)	0.78 ± 0.13	1.10 ± 0.13	1.48 ± 0.22
Other nutrients			
Dietary fats (% of energy)	35.1 ± 6.4	36.0 ± 6.4	36.3 ± 6.6
Carbohydrates (% of energy)	48.0 ± 8.6	44.9 ± 7.6	44.0 ± 7.7
Total fat (g)	62.3 ± 20.0	76.3 ± 23.9	96.7 ± 30.1
Saturated fat (g)	21.5 ± 8.3	26.3 ± 9.5	33.3 ± 12.1
Carbohydrate (g)	191.2 ± 59.6	213.9 ± 64.1	263.5 ± 83.5
Dietary fiber (g)	13.2 ± 4.8	15.6 ± 5.5	19.1 ± 7.7
Food intakes			
FnsVeg (cup-eq)	2.12 ± 1.27	2.45 ± 1.26	2.88 ± 1.65
Dairy (cup-eq)	1.02 ± 0.62	1.37 ± 0.73	1.84 ± 1.10
Red meat (oz-eq)	1.67 ± 1.22	2.24 ± 1.46	3.10 ± 1.84
Whole grains (oz-eq)	0.47 ± 0.58	0.59 ± 0.68	0.77 ± 0.87
HEI-2015 scores	56.8 ± 11.9	55.8 ± 12.1	55.6 ± 11.7
Alcohol (g per day) (median, IQR)	7.0 (2.5–15.7)	10.1 (3.5–18.9)	9.6 (3.9–21.2)
Sex (n, % male)	156 (39.1)	262 (40.6)	173 (45.8)
High school or beyond (n, %)	238 (59.7)	462 (71.5)	261 (69.0)

^1^ Expressed as means (standard error) or otherwise stated. ^2^ Low protein is defined as <75 g/day for men, <65 g/day for women; moderate protein is 75–<100 g/day for men, 65–<85 g/day for women; high protein is ≥100 g/day for men and ≥85 g/day for women. BMI, body mass index; eq, equivalent; FnsVeg, fruit, and non-starchy vegetables; kcals, kilocalories, and METs, metabolic equivalents.

**Table 2 nutrients-17-00083-t002:** Risk of impaired fasting glucose or type 2 diabetes according to categories of total, animal, and plant protein intakes in the Framingham Offspring Study.

	All subjects			Women			Men		
	Cases/N	Rate/ 1000PY	HR (95%CI) ^4^	Cases/N	Rate/ 1000PY	HR (95%CI) ^4^	Cases/N	Rate/ 1000PY	HR (95%CI) ^4^
Total protein ^1^									
Low	245/399	46.5	1 (Ref.)	132/243	37.8	1 (Ref.)	113/156	63.4	1 (Ref.)
Moderate	354/646	39.2	0.79 (0.66, 0.94)	174/384	30.4	0.77 (0.60, 0.98)	180/262	54.5	0.83 (0.64, 1.07)
High	191/378	35.1	0.69 (054, 0.87)	81/205	24.4	0.61 (0.42, 0.87)	110/173	51.8	0.82 (0.58, 1.15)
Animal protein ^2^									
Low	234/359	52.5	1 (Ref.)	87/151	42.1	1 (Ref.)	147/208	61.5	1 (Ref.)
Moderate	233/456	35.1	0.71 (0.58, 0.86)	149/331	29.5	0.61 (0.46, 0.80)	84/125	53.1	0.80 (0.60, 1.06)
High	323/608	37.3	0.68 (0.55, 0.83)	151/350	27.9	0.59 (0.43, 0.80)	172/258	53.1	0.76 (0.59, 0.99)
Plant protein ^3^									
Low	364/647	41.2	1 (Ref.)	217/438	33.8	1 (Ref.)	147/209	60.6	1 (Ref.)
Moderate	202/375	37.2	0.89 (0.74, 1.08)	109/235	29.2	0.88 (0.67, 1.15)	93/140	54.5	0.91 (0.69, 1.21)
High	224/401	41.0	0.94 (0.74, 1.20)	61/159	25.5	0.86 (0.58, 1.28)	163/242	52.9	1.01 (0.74, 1.37)

^1^ Low protein is defined as <75 g/day for men, <65 g/day for women; moderate protein is 75–<100 g/day for men, 65–<85 g/day for women; and high protein is ≥100 g/day for men and ≥85 g/day for women. ^2^ Low animal protein is defined as <55 g/day men, <40 g/day women; moderate 55–<65 g/day men, 40–<55 g/day women; and high ≥65 g/day men, ≥45 g/day women. ^3^ Low plant protein is defined as low <20 g/day, moderate 20–<25 g/day, and high ≥25 g/day. ^4^ Models adjusted for age, sex (only in All subjects), education level, physical activity, smoking cigarettes per day, height, energy intake, and weight-adjusted carbohydrate intake; the animal protein model adjusts for plant protein intake, and the plant protein model adjusts for animal protein intake. PY, person-years.

## Data Availability

Restrictions apply to the availability of these data. Data was obtained from the Framingham Heart Study and can be requested at https://framinghamheartstudy.org/fhs-for-researchers/data-available-overview/ (Accessed 25 December 2024).

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
