# Peer review of "Individual and Joint Associations Between Animal and Plant Protein Intakes with Impaired Fasting Glucose and Type 2 Diabetes in the Framingham Offspring Study"

_nutrients, 2024, doi:10.3390/nu17010083_

Round 1

Reviewer 1 Report

Comments and Suggestions for Authors

The authors provide results from a prospective cohort study, investigating the association of the intake of plant and animal protein on diabetes-related outcomes.

The overall rationale is clear.

Abstract: Please state the timeframe of data assessment for this observation period.
Discussion/Conclusion needs to be highlighted and extended in the abstract.

Introduction: There is lack of alternative explanations for the previously found associations: protein intake is expensive and requires higher SES, which itself is associated with lower health risks. High-protein diets are more common among younger persons (bodybuilders), which are healthier for dozens of reasons (confounders). On the other hand, diets with pronounced or reduced protein intake may FOLLOW (not preceed) age- and obesity-related health problems. No observational study is safe from reverse causality. The introduction must clarify, which of these explanations might fit the "epochal" settings of the FOS. In the 1990s, various dietary approaches were not as popular as today, certain dietary patterns had a different socioeconomic or motivational background.

Methods:

Fig. 1 is misleading, indicating that protein intake was assessed from Ex3 TO Ex5, while it was only done at Ex3 AND Ex5.

L. 115: please use "antidiabetic" or "anti-hyperglycemic" instead of "hypoglycemic". Hypoglycemia is not the intented goal of these drugs.

Results:
Table 1 lacks p-values for comparison.

Table 1 shows group differences in kcal intake of 20 % between low-middle and middle-high. This does not match the minimal or even contradictive differences in physical activity, sex or BMI.

Fig. 3: Please clarify the nature of the use cut-offs for HEI, physical activity and BMI. The cut-offs do not seem to be cohort medians, so they appear to be chosen arbitrarily.

Discussion and overall: The term "effect" implies causality and should be avoided.

L. 358: Please differentiate the PREVIEW project [14], an overarching project on protein intake and health outcomes, including the referenced SRMA, and the PREVIEW study [Raben A, Vestentoft PS, Brand-Miller J, Jalo E, Drummen M, Simpson L, Martinez JA, Handjieva-Darlenska T, Stratton G, Huttunen-Lenz M, Lam T, Sundvall J, Muirhead R, Poppitt S, Ritz C, Pietiläinen KH, Westerterp-Plantenga M, Taylor MA, Navas-Carretero S, Handjiev S, McNarry MA, Hansen S, Råman L, Brodie S, Silvestre MP, Adam TC, Macdonald IA, San-Cristobal R, Boyadjieva N, Mackintosh KA, Schlicht W, Liu A, Larsen TM, Fogelholm M. The PREVIEW intervention study: Results from a 3-year randomized 2 x 2 factorial multinational trial investigating the role of protein, glycaemic index and physical activity for prevention of type 2 diabetes. Diabetes Obes Metab. 2021 Feb;23(2):324-337. doi: 10.1111/dom.14219. Epub 2020 Nov 3. PMID: 33026154; PMCID: PMC8120810.], a distinct RCT on that topic.

Limitations:

Residual confounding is missing.
2 x 3 days of food record representing eight years of observation is a mere spotlight, not a highly reliable dietary assessment.
The cohort consists of mostly well educated persons, examined up to 30-40 years ago with the socioeconomic and general living conditions of that time; validity for less educated people of that time and for people of today is limited, as overall living conditions have changed immensely.

Conclusion:

L. 427: "lower" is inappropriate, as it implies causality.

In general, there is need for RCTs, not just more inconclusive cohort studies.

Author Response

Abstract: Please state the timeframe of data assessment for this observation period.

We have added the timeframe for diet assessment and the max duration of follow-up to the abstract.

Discussion/Conclusion needs to be highlighted and extended in the abstract.

We have highlighted the Conclusions heading. Unfortunately, due to word limit constraints, we are unable to expand much beyond what is given due to changes requested by other referees.

Introduction: There is lack of alternative explanations for the previously found associations: protein intake is expensive and requires higher SES, which itself is associated with lower health risks. High-protein diets are more common among younger persons (bodybuilders), which are healthier for dozens of reasons (confounders). On the other hand, diets with pronounced or reduced protein intake may FOLLOW (not preceed) age- and obesity-related health problems. No observational study is safe from reverse causality. The introduction must clarify, which of these explanations might fit the "epochal" settings of the FOS. In the 1990s, various dietary approaches were not as popular as today, certain dietary patterns had a different socioeconomic or motivational background.

We agree that high protein diets can be confounded by other factors related to physical activity and socioeconomic factors. We clarified in the introduction that studies examining the relation between protein and glucose homeostasis at a population level have controlled for these factors. In this analysis, level of education attained was used as a proxy for Socioeconomic status, and was not linearly associated with protein intake, limiting the probability that the data is biased due to this metric.

Methods:

Fig. 1 is misleading, indicating that protein intake was assessed from Ex3 TO Ex5, while it was only done at Ex3 AND Ex5.

Thank you for pointing this out. We have included language in the legend to state that arrows indicate when specific data were taken.

115: please use "antidiabetic" or "anti-hyperglycemic" instead of "hypoglycemic". Hypoglycemia is not the intented goal of these drugs.

We have made this change.

Results:
Table 1 lacks p-values for comparison.

The results in Table 1 present baseline characteristics of the study participants according to the intakes of protein. Since this table of baseline characteristics is designed to enable the reader to consider what factors may be potential confounders, and since p-values do not inform us about the potential for actual confounding, we prefer not to include them in this table. In accordance with the STROBE guidelines on reporting observational data, we have excluded tests of statistical significance from the descriptive tables (http://www.plosmedicine.org/article/info%3Adoi%2F10.1371%2Fjournal.pmed.0040297).

Table 1 shows group differences in kcal intake of 20 % between low-middle and middle-high. This does not match the minimal or even contradictive differences in physical activity, sex or BMI.

Although food records provide a more accurate dietary assessment compared to food frequency questionnaires (commonly used in most prior studies), they still have certain limitations. A notable issue is the underreporting of calorie-dense, low-nutrient foods such as cookies, candy, and similar items—an observation consistent with most epidemiological studies. We appreciate the reviewer’s comment and have addressed this limitation in the discussion section (440-442).

Fig. 3: Please clarify the nature of the use cut-offs for HEI, physical activity and BMI. The cut-offs do not seem to be cohort medians, so they appear to be chosen arbitrarily.

Our goal was to select meaningful cutoff values for these effect modifiers that would also provide sufficient power. Specifically, the values that we chose for the lowest and highest categories of HEI, PA comprised 33%  vs 66%, 40%  vs 60% of subjects at each end of the distribution, respectively. The sex-specific cut-offs for BMI (lower vs. higher BMI: <28 kg/m2 for men and <25 kg/m2 for women vs. ≥28 kg/m2 for men and ≥25 kg/m2 for women) were also based on our previous work in Offspring adults showing that cardiometabolic risk develops at a higher BMI (~28 kg/m2). It is possible that for these middle-aged men, a BMI between 25 and 28 kg/m2 reflects higher concentrations of lean mass rather than an excess of adiposity.

Chadid, S.; Kreger, B.E.; Singer, M.R.; Loring Bradlee, M.; Moore, L.L. Anthropometric Measures of Body Fat and Obesity-Related Cancer Risk: Sex-Specific Differences in Framingham Offspring Study Adults. Int J Obes 2020, 44, 601–608, doi:10.1038/s41366-020-0519-5.

We have added the cutoffs in the methods (lines 160-170) and they are also described in figure legends.

Discussion and overall: The term "effect" implies causality and should be avoided.

Thank you for the comment, we agree. We have changed the manuscript to limit the use of “effect” as a stand in for “associations” or “relations”, except for “effect modification analysis”.

  1. 358: Please differentiate the PREVIEW project [14], an overarching project on protein intake and health outcomes, including the referenced SRMA, and the PREVIEW study [Raben A, Vestentoft PS, Brand-Miller J, Jalo E, Drummen M, Simpson L, Martinez JA, Handjieva-Darlenska T, Stratton G, Huttunen-Lenz M, Lam T, Sundvall J, Muirhead R, Poppitt S, Ritz C, Pietiläinen KH, Westerterp-Plantenga M, Taylor MA, Navas-Carretero S, Handjiev S, McNarry MA, Hansen S, Råman L, Brodie S, Silvestre MP, Adam TC, Macdonald IA, San-Cristobal R, Boyadjieva N, Mackintosh KA, Schlicht W, Liu A, Larsen TM, Fogelholm M. The PREVIEW intervention study: Results from a 3-year randomized 2 x 2 factorial multinational trial investigating the role of protein, glycaemic index and physical activity for prevention of type 2 diabetes. Diabetes Obes Metab. 2021 Feb;23(2):324-337. doi: 10.1111/dom.14219. Epub 2020 Nov 3. PMID: 33026154; PMCID: PMC8120810.], a distinct RCT on that topic.

We appreciate the reviewer’s clarification on this. We have changed the language to specifically indicate that we are referencing the PREVIEW project, not the similarly named studies.

Limitations:

Residual confounding is missing.

We have added residual confounding as a limitation in the discussion (line 447).

2 x 3 days of food record representing eight years of observation is a mere spotlight, not a highly reliable dietary assessment.

The collection of dietary data in this study was careful and systematic. We have added more details of the gold standard methods used for estimating dietary intake to the discussion. As a part of the collection of the dietary data, trained senior research nutritionists debriefed each subject and determined whether the dietary intake data were likely to be reliable or not. It is worth noting that much of the underreporting in dietary studies tends to involve snacks and sweets rather than protein intake. Therefore, we believe the reported protein consumption in our study is accurate and reliable. We also think that the ~12-year period for dietary assessment is important because the large number of dietary records (~16,000) provides us with highly precise estimates of habitual long-term protein intake. This long-term measure is likely to have a more meaningful impact on diabetes risk than short-term fluctuations in protein consumption.

The cohort consists of mostly well educated persons, examined up to 30-40 years ago with the socioeconomic and general living conditions of that time; validity for less educated people of that time and for people of today is limited, as overall living conditions have changed immensely.

We appreciate the reviewer’s comment and added it to the limitations (lines 448-449).

Conclusion:

  1. 427: "lower" is inappropriate, as it implies causality.

We have rephrased the sentence (Line 454)

In general, there is need for RCTs, not just more inconclusive cohort studies.

We have added the need for clinical studies to conclusion (Line 455)

Reviewer 2 Report

Comments and Suggestions for Authors

The investigations carried out by Pickering et al can be considered for publication after the authors consider to address my suggestions, as follows:

Firstly, the authors have to rewrite the manuscript to decrease drastically the similarity index once 38% is too high.

The abstract should have no more than 250 words. See the journal guidelines.

In the abstract, conclusions and future perspectives are missing.

Improve your Introduction. A more robust background to well justify the need for your study should be provided.

Line 82: What is the Framingham Offspring Study? References are missing in the section 2.1.

The section 2.5 is too extensive. It should be reduced to the essentials.

Lines 299-310: This is not discussion and should be deleted from this section, probably the authors can move it to the Results section.

Include more studies in the Discussion section to make it more robust and compare the obtained results in these studies with yours.

Conclusions are poor. Elaborate on them and give some relevant implications, future perspectives, and the impact of your obtained results on the scientific community and population.

Author Response

The investigations carried out by Pickering et al can be considered for publication after the authors consider to address my suggestions, as follows:

Firstly, the authors have to rewrite the manuscript to decrease drastically the similarity index once 38% is too high.

We have addressed this with the editorial committee. In short the metric used for similarity here is inherently flawed. Our response to the editorial assistant was as follows, which was reviewed and approved by the editorial department.

“With regards to the duplication measures, I am struggling to interpret and identify where the majority of these duplications are being identified from.

To my reading, a significant percentage of them are due to the use of the MDPI nutrients template and accepted cookie cutter statements regarding data availability and funding statements (#2). The remainder of the duplications found in #2 are from a paper published by our group in Nutrients, so certain phrases are reused for consistency of analysis and interpretation.

Further, I have been unable to identify the source of the most highly "duplicated" readout, as this seems to be a repository of many, many research programs. Searches that include the specific language yield no results that match any one source, they are disparate and not connected, and have very general patterns that would used in the discussion of protein and weight such as ""The effects of a high protein diet on weight loss." Thus, it seems this repository would have most if not all language used in the discussion of population based studies.

Even those with very specific references, (such as #9, which is also our group) reference a very specific method or data source, that would otherwise be uninterpretable with changes to the language.

Further, I fail to see the paragraphs which you say have "some duplication in whole paragraphs," Outside those provided by "Nutrients"

Finally, the vast majority of these duplications reference something exceedingly general to the effect of "models were adjusted for sex, age", or "(95% CI 0.XX, 0.XX)" which are inescapable for population-based studies or standards of reporting.

The abstract should have no more than 250 words. See the journal guidelines.

We have shortened the abstract to 250 words (lines 11-27).

In the abstract, conclusions and future perspectives are missing.

We have added conclusions to the abstract (lines 25-27)

Improve your Introduction. A more robust background to well justify the need for your study should be provided.

In our introduction, we have characterized the uncertainty in the field regarding the association between total protein intake and measures of glucose homeostasis using not less than 10 different studies of various approaches (Lines 47-54). Further, we have highlighted other uncertainties as they relate to BMI, diet and other lifestyle factors within these studies, and in additional studies examining these more specific questions (55-72); major goals we address in our manuscript. We are unsure as what else the reviewer requires for justification of our goals stated in line 76-79 without more specific commentary.

Line 82: What is the Framingham Offspring Study? References are missing in the section 2.1.

We have added the citation for FOS (line 83).

The section 2.5 is too extensive. It should be reduced to the essentials.

We have shortened the section as much as we could. Some details for effect modification were kept as they address concerns by the other reviewer.

Lines 299-302 : This is not discussion and should be deleted from this section, probably the authors can move it to the Results section.

We assume this comment refers to the first paragraph of the discussion, as in the original submission 299-302 are contained within the results section. Regardless, the first paragraph is meant to be a summary of the major results of the paper, which is common practice.

Include more studies in the Discussion section to make it more robust and compare the obtained results in these studies with yours.

We are unclear as to what the reviewer would ask us to add and in what topic area. The discussion spanning over 20 references is dedicated to interpreting the observed results in the context of other studies examining the association with protein intake and diabetes/impaired glucose homeostasis risk.

Conclusions are poor. Elaborate on them and give some relevant implications, future perspectives, and the impact of your obtained results on the scientific community and population.

Reviewer 3 Report

Comments and Suggestions for Authors

This appears to be a revised version. Previous data on dietary protein intake and risk of subsequent diabetes has shown conflicting associations.

The authors have data on both animal protein and plant protein intake as well as the combination and over 16 years of follow up. Dietary intake data methodology has its limitations but theirs using three day records on two occasions is probably as good as one can get. They also address potential confounders. 

Their main finding is that subjects consuming more more protein especially animal protein particularly those with lower adiposity have a lower risk of incident T2DM and IFG.

They discuss the strengths and limitations of their study and other studies on this topic.

The various sections of the paper are all adequate. I have no specific issues to be further addressed

Author Response

We thank the reviewer for their time and consideration.

Round 2

Reviewer 1 Report

Comments and Suggestions for Authors

The authors have revised their manuscript following the reviewer's guidance.

Several points remain to be improved:

Introduction: Confounders such as SES, environmental factors and others CANNOT be adequately controlled in cohort studies, as these factors are either insufficiently measurable or measured, or they withstand linear adjustment for being associated in a non-linear fashion.

Results:

If table 1 is presented in order to inform about potential confounders, p values are particularly necessary to highlight confounding variables, as the ones with significant differences are mostly like the relevant ones. Thus, please include p values. Otherwise your choice of adjustment variables would seem to be arbitrary nonetheless.

Misreporting of diet: Did you exclude patients with clearly implausible dietary data? Based on which cut-offs?

Discussion:

The cohort consists of persons, examined up to 30-40 years ago with the socioeconomic and general living conditions of that time; validity for people of today is limited, as overall living conditions have changed immensely. Vegans in the 1980s are not comparable to contemporary vegans.

Conclusion:

In general, there is need for RCTs, not more inconclusive cohort studies. Epidemiological research provides more and more large data sets with finally high levels of uncertainty. They provided valuable hypotheses in the beginning, but nowadays, cohort studies provide hypotheses in all directions, based on their time point and time frame of assessment. Ancel Keys found the association for SFA intake. The same study with data collected today would find a neutral or inverse association, despite identical methodology. Twenty confirmatory cohort studies won't help us, twenty inconclusive or contradictory ones will neither.

Author Response

We appreciate the reviewer’s continued and specific commentary. Our responses are below

Several points remain to be improved:

Introduction: Confounders such as SES, environmental factors and others CANNOT be adequately controlled in cohort studies, as these factors are either insufficiently measurable or measured, or they withstand linear adjustment for being associated in a non-linear fashion.

We agree that inadequate measurement of SES factors can impact overall effect measures. We have amended in the introduction(53-54, 62-64) and limitations to address this issue as well other possible residual confounding such as time period. (457-459)

Results:

If table 1 is presented in order to inform about potential confounders, p values are particularly necessary to highlight confounding variables, as the ones with significant differences are mostly like the relevant ones. Thus, please include p values. Otherwise your choice of adjustment variables would seem to be arbitrary nonetheless.

We understand the reviewers concerns. However, Since this table of baseline characteristics is designed to enable the reader to consider what factors may be potential confounders, and since p-values do not inform us about the potential for actual confounding, we prefer not to include them in this table. Me must reiterate this decision to not report p-values for descriptive tables is based in best practices as outlined by STROBE (Strengthening the Reporting of Observational Studies in Epidemiology)  (https://journals.plos.org/plosmedicine/article?id=10.1371/journal.pmed.0040297)

Potential confounding variables were identified from relevant literature and tested following a systematic approach as described in our methods (Line 201).

Misreporting of diet: Did you exclude patients with clearly implausible dietary data? Based on which cut-offs?

Yes, we have clarified what is meant by valid dietary information to include exclusion of extreme energy intakes (Line 102-104)

Discussion:

The cohort consists of persons, examined up to 30-40 years ago with the socioeconomic and general living conditions of that time; validity for people of today is limited, as overall living conditions have changed immensely. Vegans in the 1980s are not comparable to contemporary vegans.

We appreciate the reviewers comment and have added to our limitations that dietary intakes of the past may not reflect those of today. (440-441)

Conclusion:

In general, there is need for RCTs, not more inconclusive cohort studies. Epidemiological research provides more and more large data sets with finally high levels of uncertainty. They provided valuable hypotheses in the beginning, but nowadays, cohort studies provide hypotheses in all directions, based on their time point and time frame of assessment. Ancel Keys found the association for SFA intake. The same study with data collected today would find a neutral or inverse association, despite identical methodology. Twenty confirmatory cohort studies won't help us, twenty inconclusive or contradictory ones will neither.

We thank the reviewer for their comment on this issue, and agree that RCTs are needed for more directly testing the associations and causality between these dietary factors. However the time to develop type 2 diabetes as a result of habitual behaviors like protein intake or dietary patterns is difficult to assess in clinical trials outside of tour de force studies like PREDIMED.

However, we have amended the statement to only include well-designed trials to better test the association between these factors. (445)

Reviewer 2 Report

Comments and Suggestions for Authors

I still believe the authors should decrease the similarity index. However, I leave this to the considerations that can be made by the editor.

Conclusions still need to be improved, as I previously stated.

Author Response

Conclusions still need to be improved, as I previously stated.

We have expanded the conclusions to include that higher protein diets may be a consideration for the prevention of type 2 diabetes.